# Informing ASR Treatment Practices in a Florida Aquifer through a Human Health Risk Approach

**DOI:** 10.3390/ijerph20196833

**Published:** 2023-09-26

**Authors:** Anna Gitter, Kristina D. Mena, John T. Lisle

**Affiliations:** 1Department of Epidemiology, Human Genetics and Environmental Sciences, University of Texas Health Science Center-Houston School of Public Health, Houston, TX 77030, USA; kristina.d.mena@uth.tmc.edu; 2U.S. Geological Survey, St. Petersburg Coastal and Marine Science Center, St. Petersburg, FL 33701, USA; jlisle@usgs.gov

**Keywords:** QMRA, DALYs, pathogens, aquifer, water quality, human health, aquifer storage and recovery

## Abstract

Aquifer storage and recovery (ASR) can augment water supplies and hydrologic flows under varying climatic conditions. However, imposing drinking water regulations on ASR practices, including pre-treatment before injection into the aquifer, remains arguable. Microbial inactivation data—*Escherichia coli*, *Pseudomonas aeruginosa*, poliovirus type 1 and *Cryptosporidium parvum*—were used in a human health risk assessment to identify how the storage time of recharged water in the Floridan Aquifer enhances pathogen inactivation, thereby mitigating the human health risks associated with ingestion. We used a quantitative microbial risk assessment to evaluate the risks for a gastrointestinal infection (GI) and the associated disability-adjusted life years (DALYs) per person per year. The risk of developing a GI infection for drinking water no longer exceeded the suggested annual risk threshold (1 × 10^−4^) by days 31, 1, 52 and 80 for each pathogen, respectively. DALYs per person per year no longer exceeded the World Health Organization threshold (1 × 10^−6^) by days 27, <1, 43 and 72. In summary, storage time in the aquifer yields a significant reduction in health risk. The findings emphasize that considering microbial inactivation, caused by storage time and geochemical conditions within ASR storage zones, is critical for recharge water treatment processes.

## 1. Introduction

The critical need to manage limited and often unpredictable water resources, affected by climate change, population growth and agricultural practices, has resulted in an array of water reuse and recovery applications to address and respond to these increased demands. Aquifer storage and recovery (ASR) is a type of managed aquifer recharge (MAR) technology that injects treated surface water into an aquifer storage zone for later recovery for surface applications [1]. The application of ASR for restoring and maintaining hydrologic flow through an ecosystem is currently being implemented as part of the Comprehensive Everglades Restoration Plan (CERP) [2,3]. ASR facilities associated with CERP will recharge treated surface water into specific zones of the Floridan Aquifer during climatic “wet” periods (e.g., rainy season, post-hurricane) when hydraulic flow through the Everglades is sufficient. During periods when this hydraulic flow decreases below a critical rate, the stored surface water will be recovered and discharged into water bodies located in the headwaters of the Everglades (e.g., Lake Okeechobee) to artificially augment the volume of water flowing through the Everglades at rates that support ecosystem health. 

The surface water recharged into aquifer storage zones at ASR facilities must be monitored at the above-ground point of injection for specific constituents, including microbial contaminants, as described in the primary drinking water standards of the Safe Drinking Water Act (SDWA) [4,5,6,7,8] and implemented by the Florida Department of Environmental Protection [9]. Interestingly, there is no regulatory requirement to monitor the water quality of the recovered recharge water at the point of discharge. While the SDWA requires routine sampling at points of customer consumption (most often a residential faucet), the recovered recharged water at CERP-related ASR facilities is discharged into lakes, streams and/or wetlands, within the Everglades watershed.

By not sampling the recovered and discharged ASR water for the same microbiological constituents as required for the surface water prior to recharge, the influence of time and geochemical conditions of the storage zone on the survival of these microorganisms is overlooked. In fact, fecal coliforms, *Escherichia coli* (*E. coli*), bacteriophages (e.g., MS2) and eukaryotic viruses (e.g., adenovirus, rotavirus) and encysted protozoans (e.g., *Cryptosporidium* sp., *Giardia* sp.) have been shown to be inactivated during the storage of recharged surface water in aquifer zones [10,11,12,13,14], including the Floridan Aquifer [15,16]. 

Utilizing a human health risk framework to estimate and interpret health risks associated with environmental hazards and policies has been extensively used for recreational, drinking and groundwater management [10,11,17,18,19]. This mathematical approach utilizes four phases (hazard identification, exposure assessment, dose-response assessment and risk characterization) to estimate the risk of infection associated with exposure to microbial contaminants [20,21]. Human health risk assessment, specifically quantitative microbial risk assessment (QMRA), can be used to inform how environmental conditions and regulatory policies may influence these health risks. Previous studies have evaluated the efficacy of MAR practices for pathogen inactivation through QMRA [10,11]. Toze et al. [11] determined that recovered water at an MAR research site that underwent infiltration and aquifer passage did not meet the Australian Guidelines for recycled water, therefore emphasizing the importance of utilizing local hydrogeological data to inform MAR practices. Page et al. [10] determined that incorporating an aquifer barrier for the subsurface treatment of urban stormwater assisted in pathogen inactivation and mitigating health risks (as demonstrated through disability-adjusted life years (DALYs) < 1 × 10^−6^). 

In this paper, we describe the application of QMRA as an exercise to translate pathogen reduction during the storage phase in aquifer environments under ASR processes to estimate health risk outputs. The human health risk assessment approach presented in the study assesses the health risks associated with two specific exposure scenarios: ingestion of water from drinking and while swimming. We chose to use both scenarios due to the regulatory requirements for surface water to be treated to drinking water standards at ASR facilities in Florida [9] (though the recovered water will not be a direct or indirect potable source) and that the recovered water will be discharged into water bodies, which are routinely used for recreational activities. This approach can inform policy regarding the value of pathogen inactivation during storage cycles at ASR facilities in the context of meeting regulatory standards and the critical need to incorporate environmental conditions into treatment trains, regulatory criteria and decision making. This study utilizes previously published pathogen inactivation rates from studies in the Upper Floridan (UF) and Avon Park Permeable Zone (APPZ) zones within the Floridan Aquifer System located in central-to-south Florida [15,16] to inform pathogen inactivation and human health risks over time in the respective ASR storage zones. 

## 2. Materials and Methods

### 2.1. Groundwater Conditions of the Study Area

#### 2.1.1. Hydrogeology

All inactivation rate data were generated from experiments conducted in two non-potable aquifer zones within the Floridan Aquifer System in central-to-south Florida: Upper Floridan (UF) and Avon Park Permeable Zone (APPZ) [15,16,22]. The depth of each aquifer zone from which the groundwater was accessed and associated information for each of the sample sites are provided in Table 1 and Figure 1. 

The geologic framework of the Floridan Aquifer System in this region of Florida consists of a thick sequence of predominantly marine carbonates that were deposited during the middle to late Eocene (Avon Park Formation and Ocala Limestone) and Oligocene (Suwannee Limestone and basal Arcadia Formation) [23]. In order of oldest to youngest, the formations are the Avon Park Formation, Ocala Limestone, Suwannee Limestone and the basal unit of the Arcadia Formation [23]. The APPZ zone is included entirely within the Avon Park Formation, which consists of fine-grained micritic to fossiliferous limestone, dolomitic limestone and dolostone or dolomite, and it is frequently fractured. Fine- to medium-grained calcarenite is also present, though intermittently [23]. The UF zone is included within either the Ocala Limestone or the Suwannee Limestone formations. The Ocala Limestone is also fossiliferous and consists of micritic or chalky limestone, calcarenitic limestone and coquinoid limestone. The Suwannee Limestone is also fossiliferous with medium-grained calcarenite and an intermittent zone of quartz sand and phosphate mineral grains [23]. Hydraulic conductivities ranged between 1.5–15.2 m day^−1^ and 24.4–1524 m day^−1^ for the wells accessed during the cited studies for the UF and APPZ, respectively [23]. All ASR recharge and storage zones are below zones of potable groundwater sources, physically isolated from those sources by an upper confining unit of interlayered clays, silts and fine sands (i.e., Hawthorn Formation) and is not impacted by meteoric or surface water [24].

#### 2.1.2. Groundwater Chemistry

The water quality data summaries from the groundwater wells accessed during the inactivation studies are listed in Table 2. All groundwater zones are anaerobic and contain no ${NO}_{3}$ or ${NO}_{2}$ [15,16,22].

#### 2.1.3. Inactivation Data for *E. coli*, *Pseudomonas aeruginosa*, Poliovirus Type 1 and *Cryptosporidium parvum*

Previous microbial inactivation studies in the aquifer zones described in Section 2.1.1 and Section 2.1.2 determined that the following microorganisms—*E. coli*, *Pseudomonas aeruginosa* (*P. aeruginosa*), poliovirus type 1 (PV1) and *Cryptosporidium parvum* (*C. parvum*)—followed bi-phasic (*E. coli* and *P. aeruginosa*), log-linear (PV1) and Weibull (*C. parvum*) inactivation models [15,16]. The inactivation model parameters are summarized in Table 3. Descriptions of study design, microorganism enumeration and statistical analyses are not reported in this study and can be found in the published literature [15,16,22]. 

### 2.2. Human Health Risk Assessment

#### 2.2.1. Quantitative Microbial Risk Assessment 

The QMRA framework has been extensively applied to evaluate human health risks associated with exposure to microbial contaminants in different water systems (e.g., groundwater, drinking water, recreational water, water reuse, etc.) [10,11,12,17,18,19,25,26]. The framework aims to characterize the contaminant, evaluate the scenario by which an individual is exposed, incorporate a dose–response model representative of the pathogens of interest and estimate the potential risk of infection and illness. Inputs (Table 4) utilized in the QMRA model included the following: (a) ingestion volume (from drinking water and incidental ingestion while swimming) and (b) pathogen inactivation rates. Human health risks were evaluated in context of a suggested annual risk threshold for drinking water (1 infection per 10,000 individuals or 1 × 10^−4^), given that the water is monitored to meet primary drinking water standards [9,27]. The disability-adjusted life years (DALYs) associated with the annual risk of illness for each pathogen were estimated and compared to the WHO DALY risk threshold (1 case per 1,000,000 individuals or 1 × 10^−6^) [28]. DALY estimations required the following inputs (Table 5): disease burden per case, susceptibility, disease severity and duration (years). 

The study also assumes a closed system for the risk assessment model, in which water that is retrieved from the surface is partially treated via rapid sand filtration and in-line UV disinfection, then injected into the aquifer zone and resides there without assumptions for dilution and no additional water injections or withdrawals. After a specified amount of time, the water is then recovered from the aquifer. The model assesses the human health risks associated with ingestion of that recovered water. 

#### 2.2.2. Hazard Identification

The microorganisms of interest for this human health risk assessment were selected due to their regulatory significance for SDWA standards and treatment [20,21,29]. These microorganisms—*E. coli* (fecal indicator bacterium and opportunistic pathogen), *P. aeruginosa* (opportunistic bacterial pathogen), PV1 (human enterovirus) and *C. parvum* (encysted eukaryotic protozoan)—are representative of indicators and pathogens that are a significant concern for public health and groundwater. Though PV1 is not as protective as a viral surrogate as other enteric viruses (i.e., norovirus, adenovirus), conservative assumptions are utilized in this work to protectively estimate the overall impact that storage time in the aquifer can have on pathogen concentrations and, therefore, estimated human health risks. Moreover, these four microorganisms have varying survival rates in the anaerobic and reduced groundwater environments [15,16], therefore representing different microbial groups that could be of public health concern.

#### 2.2.3. Exposure Assessment

Given that the average storage time for recharged surface water from CERP-related ASR facilities is 3–6 months [15] and the intended use of the recovered water is for environmental flows, the likelihood of this water being used for drinking or directly ingested via recreational activities (e.g., swimming) is unlikely. However, the aim of this risk assessment is to identify the decrease in potential health risks associated with the recovered water after prolonged time in the aquifer storage zones. 

The microorganism concentrations considered in the exposure assessment were retrieved from environmental studies that enumerated fecal indicator bacteria and pathogen concentrations in Florida surface water targeted for ASR source waters [30]. No dilution or additional treatment/disinfection was incorporated into the risk assessment. Ingestion volumes for both drinking water and swimming are point estimates, 926 mL and 32 mL, respectively [31,32]. The dose formula includes the inactivation rate for each microorganism (Table 3 and Equation (1)). Parameters included to calculate the dose in the exposure assessment are described in Table 3. This study does not incorporate secondary transmission or immunity when evaluating human health risks [33].

**Table 3 ijerph-20-06833-t003:** Parameters included in the exposure assessment to estimate the microorganism dose.

Model Parameters	Data Type	Model Values			References
*N* _0_	Microorganisms and Concentrations at Time Zero				
	*E. coli*	3 CFU/mL			[30]
	*P. aeruginosa*	3 CFU/mL			Assumption
	PV1	1.4 × 10^−4^ PFU/mL			[30]
	*C. parvum*	0.20 oocysts/mL			[30]
*V*	Volumes Consumed				
	Drinking	926 mL			[31]
	Swimming	32 mL			[32]
*I*	Microbial Inactivation				
	Microorganism	Best Fit Model	Model Rate Constants		
	*E. coli*	Biphasic	*k* _1_	−0.4878 log_10_/day	[15,16]
			*k* _2_	−0.0118 log_10_/day	
	*P. aeruginosa*	Biphasic	*k* _1_	−0.4480 log_10_/day	[15,16]
			*k* _2_	−0.0052 log_10_/day	
	*PV1*	Log-linear	k*_max_*	−0.1900 log_10_/day	[15,16]
	*C. parvum*	Weibull	*δ*	5.04 days for 1.0 log_10_ reduction	[15,16]
			*p*	0.69	

*N*_0_ = microorganism concentration at time zero; *V* = volume consumed; *I* = microbial inactivation rate data; *p* = inactivation curve shape parameter value; CFU = colony forming units; PFU = Plaque forming units.

The dose formula is the product of the concentration of the pathogen in the water ($N_{0}$) (Table 3), the pathogen inactivation rate (*I*) (Table 3) and the assumed volume of water ingested (*V*) (Table 3) in the exposure scenario [20,21] (Equation (1)).
(1)$$Dose=N_{0}\times I\times V$$

#### 2.2.4. Dose–Response Assessment

The probability of infection following exposure to each of the pathogens was estimated using dose–response models retrieved from the published literature [34,35,36,37]. While general *E. coli* was utilized in a previous microbial inactivation study [16], we assumed that a portion of *E. coli* would be pathogenic. A ratio of 1:0.08 of general *E. coli* to pathogenic *E. coli* was assumed to adjust the dose of the pathogen [20]. All microorganisms, except for (pathogenic) *E. coli* O157:H7, are best fit by the exponential dose–response function (Equation (2)), which is described by the parameters for survival probability (*k*) and dose (*d*) (Equation (2)). *E. coli* O157:H7 is best fit by the Beta–Poisson dose–response model, which includes the parameters α, survival probability (β) and dose (*d*) (*N*_0_ in Table 3) (Equation (3)). The parameters incorporated in the dose–response models are described in Table 4.
(2)$$P\left(infection\right)=1-{exp}^{\left(-k\times d\right)}$$
(3)$$P\left(infection\right)=1-{\left(1+\left(\frac{d}{\beta }\right)\right)}^{-\alpha }$$

**Table 4 ijerph-20-06833-t004:** Dose–response parameters utilized to estimate the probability of infection.

Microorganism	Probability of Infection Parameters	References
*Pathogenic E. coli*	α = 0.248, β = 48.8	[35]
*P. aeruginosa*	k = 2.91 × 10^−9^	[37]
PV1	*k* = 0.491	[34]
*C. parvum*	*k* = 0.09	[36]

#### 2.2.5. Risk of Infection Estimates

The risk of a gastrointestinal (GI) infection (under both scenarios) was estimated using Equation (2). An annual risk of infection [20,27], assuming up to n = 30 exposures to the recovered water, was estimated. The annual infection risk was estimated using Equation (4). The number of exposure events (n = 30), given that this water is not intended for drinking water purposes, is an extremely conservative exposure frequency to assess how these residual health risks decrease following pathogen inactivation in the storage zone. For the DALYs estimations, it was assumed that every infection would result in illness. The suggested annual risk threshold, 1 × 10^−4^, is used for both exposures. While the U.S. EPA risk threshold for primary contact recreation is significantly greater, 3.6 × 10^−2^, and is a per event risk threshold, a more conservative approach when interpreting risk was applied in this study, especially given that the ASR management practices will require surface water, prior to recharge, to be treated to potable water standards [9].
(4)$$P\left(annual\;infection\right)=1-{\left(1-P\left(infection\right)\right)}^{30}$$

#### 2.2.6. DALYs per Case Estimations

The disability-adjusted life years (DALYs) metric was utilized to estimate the impact of disability (associated with illness) for public health [38]. DALYs permit health risks to be compared and risk management options to be evaluated for different public health hazards [39], all while quantifying the burden of the disease [40]. Conservative DALY estimates were developed for each representative pathogen using the QMRA risk of illness estimates (assuming each infection results in an illness), disease severity, duration and susceptibility of the population. When appropriate, the life expectancy of a Florida adult—77.1 years—was used [41]. Given the range of health outcomes associated with each pathogen, the following DALYs were calculated: pathogenic *E. coli* gastroenteritis, adenovirus gastroenteritis (adapted from PV1 due to the limited prevalence of the virus in the environment), *P. aeruginosa* gastroenteritis and cryptosporidiosis (Table 5). DALYs were a product of the annual risk of illness for that specific pathogen and the estimated DALYs per case. Additional information regarding how the DALYs per case were estimated is described in the Appendix A. 

**Table 5 ijerph-20-06833-t005:** Disability-adjusted life years (DALYs) per case parameters.

Pathogen	DALYs per Case	Description	References
Pathogenic *E. coli*	2.98 × 10^−3^	100% susceptible	[42,43]
*P. aeruginosa*	9.00 × 10^−4^	100% susceptible Assuming only gastrointestinal infection and no death	[37,44]
Adenovirus (adapted from PV1)	1.56 × 10^−3^	100% susceptible	[45]
*Cryptosporidium* spp.	3.22 × 10^−3^	100% susceptible	[43,46]

Health risks characterized as DALYs (or annual health burdens) for each pathogen are a product of the probability of illness ($P_{illness}$), as estimated with the QMRA, the susceptible fraction of the population to the disease (S) (which is assumed to be 100% for all pathogens) and the DALYs per case for mild, moderate, severe and fatal (when appropriate) gastroenteritis (B) (Equation (5)). DALYs per case estimations can be found in the Appendix A.
(5)$${DALYs}_{Pathogen}=\Sigma \;(P_{illness}\times S\times B)$$

## 3. Results

### 3.1. QMRA Risk of Infection Estimates

#### 3.1.1. Human Health Risks Associated with Drinking Water

Conservatively assuming that recovered water is untreated (except for the ASR storage phase barrier) and used directly for drinking water, the annual risk of infection for each pathogen was estimated (Figure 2). *P. aeruginosa* only exceeded the suggested annual risk of infection threshold (1 infection per 10,000 individuals) on day 0, with estimated infection risks of 2.42 × 10^−4^. Pathogenic *E. coli* initially had an elevated health risk comparable to both PV1 and *C. parvum*, yet the health risks associated with both *P. aeruginosa* and pathogenic *E. coli* declined quickly, with pathogenic *E. coli* experiencing a faster inactivation rate and, therefore, lower health risk than *P. aeruginosa*. While pathogenic *E. coli*, PV1 and *C. parvum* exceeded the risk threshold past day 1, the annual risk of infection for these pathogens decreased by four orders of magnitude by days 31, 52 and 80, respectively. The residual risk for each of the four pathogens eventually decreased to less than 2.01 × 10^−8^ by day 150. The significant and rapid decrease in the risk of infection associated with *P. aeruginosa* is likely due to the lower estimated dose of pathogen, resulting in reduced infection risks. 

#### 3.1.2. Human Health Risks Associated with Ingestion of Recreational Water While Swimming

The human health risks associated with exposure to water while swimming parallel the estimated health risks under the drinking water scenario (Figure 3). The only variable that changed between these two scenarios is the ingestion volume, in which a significantly smaller volume of water is expected to be swallowed (32 mL) while swimming compared to drinking (926 mL). Under this scenario, it is expected that the estimated health risk for a GI infection—for pathogenic *E. coli*, PV1 and *C. parvum*—would decrease by four orders of magnitude within 19, 35 and 56 days, respectively. The estimated health risks for *P. aeruginosa* never exceed the risk threshold. The residual risk for each of the four pathogens is estimated to be below 6.95 × 10^−10^ by day 150. 

### 3.2. Disease Burden per Person per Year (through DALY Estimations)

#### 3.2.1. DALYs Associated with Drinking Water Exposure

Estimated DALYs per person per year associated with gastroenteritis were compared to the acceptable threshold of 1 × 10^−6^ DALYs [28]. Under the drinking water scenario, pathogenic *E. coli*, adenovirus (representative for PV1) and *Cryptosporidium* spp. (representative for *C. parvum*) were found to exceed the risk threshold on day 0 (Figure 4). However, within 73 days, all three pathogens (pathogenic *E. coli* (27 days), adenovirus (43 days) and *Cryptosporidium* spp. (72 days)) no longer exceeded this benchmark. The DALYs associated with gastroenteritis for *P. aeruginosa* were found to be lower than the other pathogens and did not exceed the DALYs threshold. 

#### 3.2.2. DALYs Associated with Recreational Water Exposure

The decreasing trend in DALYs per person per year for swimming was similar to that of drinking water. While pathogenic *E. coli*, adenovirus and *Cryptosporidium* spp. slightly exceeded the DALYs threshold, within 49 days, the threshold was no longer exceeded (Figure 5). By day 15, pathogenic *E. coli* was estimated to decrease by four orders of magnitude. Adenovirus decreased by two orders of magnitude by day 25, and *Cryptosporidium* spp. decreased by four orders of magnitude by day 49. The differences in inactivation rates across all four pathogens indicated that by day 150, pathogenic *E. coli*, *P. aeruginosa* and adenovirus had an estimated DALYs per person per year of ≥1 × 10^−15^. *P. aeruginosa* did not exceed the DALYs threshold on day 0.

## 4. Discussion

Above-ground pre-treatment requirements for surface water to meet or exceed SDWA regulatory standards at ASR facilities in Florida pose significant increases in economic and financial obligations for the operation and maintenance of those facilities, relative to the existing requirements (e.g., rapid sand filtration and UV disinfection). However, the results from this study indicate that storage time in the aquifer and geochemical conditions in those storage zones, such as those cited in this study, are effective for the inactivation of pathogens over relatively short storage time periods. Accordingly, the storage time of recharged surface water in selected zones of the Floridan Aquifer System (i.e., UF and APPZ) could be a cost-effective alternative or supplement to the current and proposed above-ground pre-treatments. For example, a storage time of approximately 80 days of storage in the UF or APPZ would result in adequate pathogen inactivation to achieve reduced residual health risks that would meet the suggested annual risk of GI infection (1 × 10^−4^) and WHO DALY (1 × 10^−6^) recommended thresholds, as demonstrated through QMRA, for all of the pathogens represented in this study. 

### 4.1. Pathogen Inactivation in MAR/ASR Systems

Similar studies evaluating MAR systems have also identified that different groundwater characteristics can impact the inactivation rates of pathogens and should be considered in any planned MAR/ASR system [10,11,12]. In support of this proposal for ASR operations in Florida, Page et al. [10] evaluated the aquifer as a treatment zone for urban stormwater and identified a 1 to >6 ${log}_{10}$ pathogen (*Campylobacter*, *Cryptosporidium* and rotavirus) inactivation rate, therefore meeting the Australian health risk guidelines. As shown in the present study, QMRA translates the log reduction of pathogens to reductions in health risks due to different types of treatment zones and can assist in determining the types of treatments that result in acceptable health risks. Additionally, a study evaluating MAR treatment applicability for irrigation through QMRA identified that, for some regions using MAR irrigation management practices, the aquifer treatment zone was the most critical barrier for water quality treatment and pathogen inactivation [47].

### 4.2. Microbial Health Risks Associated with ASR 

QMRA has been identified as a suitable tool to evaluate microbial risks in water systems, such as for MAR/ASR water management practices [10,11,12,13,48]. This risk assessment framework was utilized to estimate residual health risks for a gastrointestinal infection across time, in the context of storage time in an aquifer zone as a treatment step for the inactivation of microorganisms of public health concern. In approximately 80 days, the residual health risks associated with exposure to these microorganisms, which experience inactivation in the groundwater environment, were less than the suggested annual risk threshold by at least four orders of magnitude. While the viral (PV1) and protozoan (*C. parvum*) pathogens had slower inactivation rates than the bacterial microorganisms (*E. coli* and *P. aeruginosa*), the health risks posed by potential exposure to these pathogens would still meet the tolerable risks suggested by public health guidelines.

The use of deterministic data in the risk assessment likely overestimates the potential human health risks (infection and DALYs), therefore serving as a conservative preliminary approach to evaluate the environmental inactivation of pathogens in the UF and APPZ zones within Floridan Aquifer System in this area of Florida. The availability and use of site-specific pathogen inactivation rates incorporated into this risk assessment framework further emphasize not only the value of QMRA for the water industry but, specifically, that this work can be incorporated into ASR plans and policies by quantifying the effectiveness of aquifer storage in reducing pathogen levels and health risks. Additionally, this risk assessment identified that storage time in the aquifer is a key factor in ensuring effective microbial inactivation and, therefore, mitigated residual health risks, which aligns with findings in other ASR-QMRA work [11]. 

When interpreting the DALYs per person per year associated with each of the pathogens, the health risks paralleled the findings from the QMRA. For gastroenteritis, the calculated DALYs per case were comparable for pathogenic *E. coli* (2.98 × 10^−3^), adenovirus (1.56 × 10^−3^) and *Cryptosporidium* spp. (3.22 × 10^−3^). *P. aeruginosa* DALYs per case (9.0 × 10^−4^) was one order of magnitude lower. A previous ASR study determined that a residence time of 70 days was insufficient for adequate pathogen inactivation (enteric pathogens: *Campylobacter*, *Cryptosporidium* and rotavirus) [11]. Another study evaluated the risks associated with the same three pathogens in an aquifer storage transfer and recovery system and found their DALYs estimations to be below the WHO threshold, with an assumed subsurface storage time of 241 days [10]. In the present study, each pathogen had estimated DALYs per person per year that no longer exceeded the WHO DALYs threshold (1 × 10^−6^) by day 73. 

The estimated DALYs incorporated into this study utilized the best available information. Metrics regarding gastroenteritis (especially associated with specific pathogens) are limited in the United States due to under-reporting or lack of diagnoses [29,49]. Many people experiencing gastrointestinal symptoms do not seek medical attention and, if they do, most often the source and causative agent are not identified. Further, many waterborne illnesses are not reportable in the U.S. so if water is determined to be the pathogen source, it is unlikely to be included in prevalence data for diarrheal illnesses. When possible, the parameters utilized for DALY estimations incorporate assumptions that would be suitable for the location of interest (e.g., average life expectancy of an adult in Florida, age range experiencing the greatest prevalence of death associated with diarrhea, etc.). 

### 4.3. Limitations

The findings presented in this study have potential to inform ASR practices but should be evaluated in the context of their limitations. The QMRA conducted incorporated deterministic values, which inhibits the consideration of the variability in parameter values. This study describes a QMRA model used to assess how pathogen inactivation in the environment correlates to estimated human health risks and, therefore, should be considered risk approximations that can continue to be revised as more relevant data become available. The microbial inactivation rates utilized in this risk assessment are specific to the environmental conditions of the ASR storage zones assessed in this study [15,16]. However, these inactivation rates are an estimation of the best available data and may not entirely capture all microbial inactivation dynamics and factors occurring in these aquifer zones. 

The human health risk outputs estimated in this study can only be assumed to be as good as the variables that were utilized to conduct the risk assessment [8]. Assumptions regarding dilution, hydrogeological conditions, microbial concentrations and exposure factors can be refined in future iterations of the model. There is also the concern if storage time has to be shortened for recharged water due to management needs. A shorter storage time (less than 80 days) could result in microbiological water quality not meeting the necessary reduction in pathogen concentrations, which would pose an unnecessary risk for human health if the water was used for drinking or recreational water. While this is a valid concern, the microbial inactivation occurring in the aquifer storage zone should not be dismissed, nor that the recovered water is not intended for drinking water purposes. The value of this QMRA model is that it shows significant decreases in quantified human health risks that do not exceed suggested health risk thresholds. This study identifies the critical importance that storage time in combination with geochemical conditions in ASR storage zones in Florida have on pathogen concentrations and, therefore, health risks. 

## 5. Conclusions

QMRA applications in the water industry translate water quality data to human health impact by quantifying the number of infections that can be prevented due to a treatment process or water management practice. In lieu of costly and logistically challenging epidemiological studies, QMRA measures a body of water as an acceptable source for human use through a range of assumptions related to contaminant type, concentration and virulence based on potential exposure dynamics. The inclusion of QMRA as part of an ASR management plan can contribute to best practices that foster a community’s water and economic sustainability. Under the assumptions utilized in this study, a storage time of at least 80 days in the UF or APPZ storage zones can influence pathogen inactivation at rates that would reduce human health risks (for a GI infection) by at least four orders of magnitude, resulting in estimated risks that no longer exceed suggested annual risk and DALY thresholds. The QMRA framework described in this study emphasizes the critical role that storage time in specific ASR storage zones has on pathogen inactivation, thereby helping to mitigate associated human health risks while complementing current water treatment practices.

## Figures and Tables

**Figure 1 ijerph-20-06833-f001:**
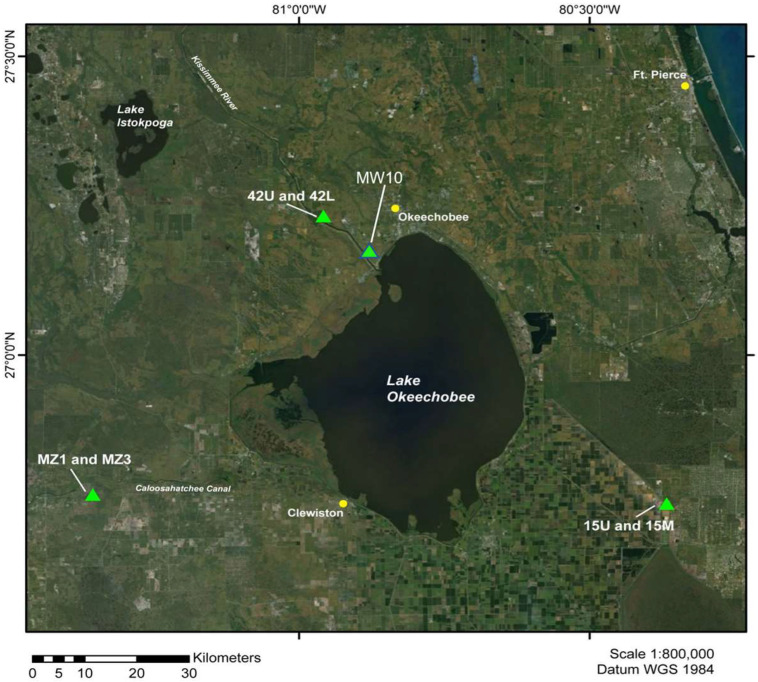
Locations of the aquifer sampling sites.

**Figure 2 ijerph-20-06833-f002:**
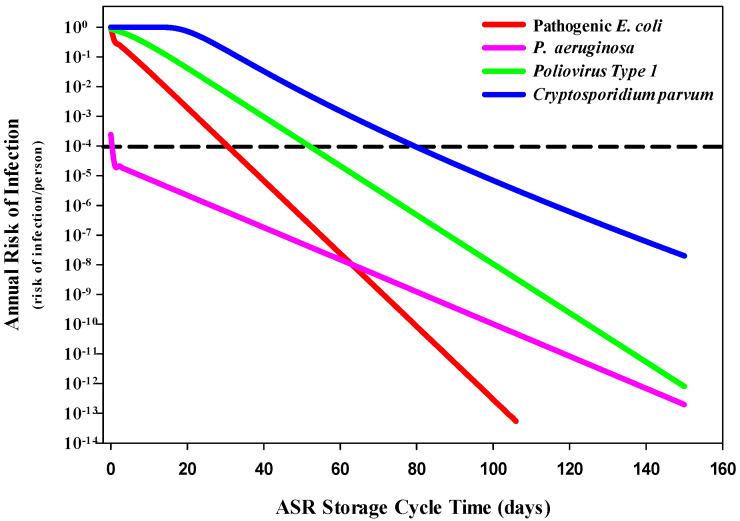
Decrease in the risk of infection associated with storage time and microbial inactivation in the groundwater system for drinking water exposure. The suggested annual risk of infection threshold (dashed line) of 1 infection per 10,000 individuals (1 × 10^−4^) was used as a reference when evaluating environmental conditions and health risks.

**Figure 3 ijerph-20-06833-f003:**
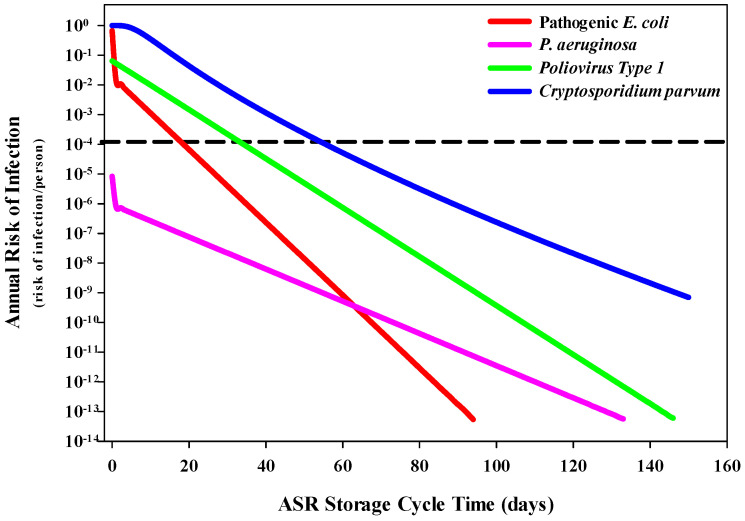
Decrease in the risk of infection associated with storage time and microbial inactivation in the groundwater system for swimming exposure. The suggested annual risk of infection threshold (dashed line) of 1 infection per 10,000 individuals (1 × 10^−4^) was used as a reference when evaluating environmental conditions and health risks.

**Figure 4 ijerph-20-06833-f004:**
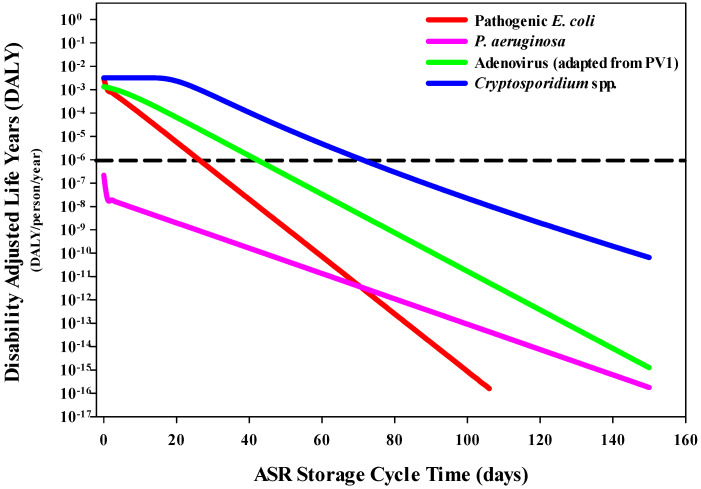
Decreasing trends in the DALYs per person per year associated with ingestion of pathogens in water that would be used for drinking. The WHO DALYs threshold (dashed line) of 1 × 10^−6^ DALYs per person per year was utilized as the reference threshold.

**Figure 5 ijerph-20-06833-f005:**
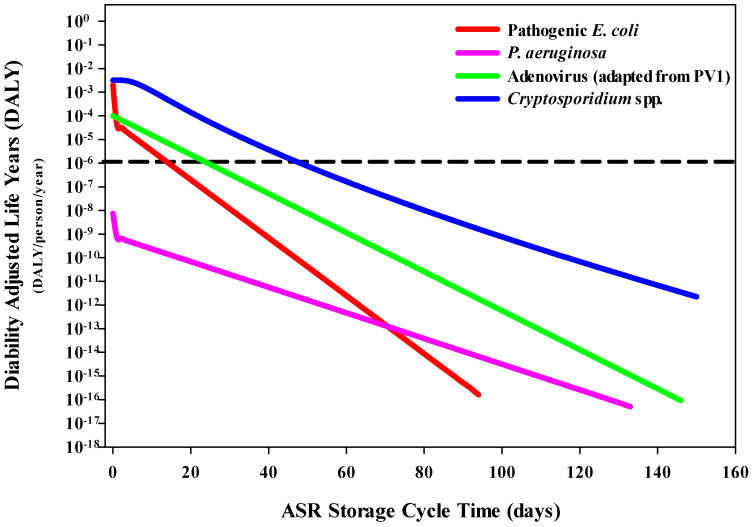
Decreasing trends in the DALYs per person per year associated with ingestion of pathogens in water while swimming. The WHO DALYs threshold (dashed line) of 1 × 10^−6^ DALYs per person per year was utilized as the reference threshold.

**Table 1 ijerph-20-06833-t001:** Description of sampling sites in the Upper Floridan and Avon Park Permeable Zone.

Well Designation	Location		Aquifer Zone	Production Interval
	Latitude	Longitude		(mbls) ^3^
MZ1	26°45′11.42″	−81°21′17.72″	UF ^1^	204–255
MZ3			APPZ ^2^	503–536
42U	27°13′11.16″	−80°57′21.98″	UF	171–317
42L			APPZ	399–469
15U	26°44′16.08″	−80°21′48.68″	UF	277–349
15M			APPZ	427–483
MW10	27°09′17.30″	−80°52′27.40″	UF	174–268

^1^ UF, Upper Floridan; ^2^ APPZ, Avon Park Permeable Zone; ^3^ mbls, meters below land surface.

**Table 2 ijerph-20-06833-t002:** Sample site water quality data.

		Well Designations						
Parameter	Units	MZ1	MZ3	42U	42L	15U	15M	MW10
Temperature	°C	28.7	27.8	28.2	28.5	27.9	28.0	25.6
Specific conductance	mS/cm	3.146	27.98	1.029	6.044	5.876	5.009	1.270
Total dissolved solids	g/L	2.045	18.19	0.669	3.928	3.819	3.255	0.728
pH		8.02	7.38	8.04	7.61	7.60	7.64	7.89
ORP	mV	−312	−309	−338	−351	−355	−365	−260
Calcium	mg/L	80.0	550.0	44.0	200.0	120.0	110.0	46.4
Chloride	mg/L	640.0	9700.0	160.0	1600.0	1600.0	1300.0	232.6
Fluoride	mg/L	0.78	BDL ^1^	0.57	0.29	0.97	1.10	0.53
Magnesium	mg/L	75.0	650.0	33.0	140.0	130.0	120.0	36.5
Potassium	mg/L	24.0	230.0	5.5	40.0	36.0	29.0	7.3
Silica	mg/L	9.8	9.1	14.0	12.0	13.0	13.0	8.2
Sodium	mg/L	440.0	4700.0	98.0	800.0	890.0	740.0	137.4
Manganese	mg/L	0.013	0.035	0.007	0.006	0.011	0.010	0.045
Iron (total)	mg/L	0.17	0.22	0.12	0.20	0.34	0.40	0.09
Ammonium	mg/L	0.19	0.28	0.20	0.26	0.44	0.33	0.22
Sulfate	mg/L	380.0	1800.0	180.0	510.0	450.0	370.0	184.6
Sulfide	mg/L	2.1	1.6	1.4	1.6	3.7	4.2	1.1
Dissolved organic carbon	mg/L	1.2	1.1	1.1	1.2	1.7	1.9	1.4

^1^ BDL, below detection limit.

## Data Availability

No new data for the generation of the QMRA models were created or analyzed in this study. Data sharing is not applicable to this article.

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
