# Peer review of "Informing ASR Treatment Practices in a Florida Aquifer through a Human Health Risk Approach"

_ijerph, 2023, doi:10.3390/ijerph20196833_

Round 1

Reviewer 1 Report

The manuscript entitled ‘Informing ASR Treatment Practices in a Florida Aquifer through a Human Health Risk Approach’ highlights the utilization of microbial inactivation data for Escherichia coli, Pseudomonas aeruginosa, poliovirus type 1 and Cryptosporidium parvum in order to investigate the effect of geochemistry and storage time in the Floridan Aquifer enhances pathogen decay. the manuscript is well written and presented and the obtained results are interesting and promising. Kindly find my comments below:

1-      It is preferable to add the links along with the references and not in the introduction part

2-      In part 2.1.3, write the bacteria names in italic

3-      In line 157, the authors mentioned that the four microorganisms have varying survival rates in the anaerobic and reduced groundwater environments. Can you specify the survival rate for each microorganism?

4-      Concerning equation (2), what does k stand for?

5-      The authors mentioned poliovirus type 1, but it is not shown in the figures.

6-      Did the authors do a wide microbiological test in order to find out the presence of additional types of bacteria (e.g. Shigella and Salmonella)

7-      The authors suggest that eliminating water treatment before injection for economic reasons, but what if sometimes the aquifer water had to be released earlier than the microorganisms decay period?   

Reviewer 2 Report

The authors address an interesting and important question in this manuscript.  I appreciate the perspective and the use of risk assessment tools to consider this issue and to add context for decision makers.  I have a number of specific comments and questions as indicated below. I hope these comments be will helpful and the authors will find constructive.

Line 21 – I find this description lacking – “The risk of a GI infection decreased” because I am not sure what pathogens you evaluated in your assessment.  Some clarity is needed in the abstract for the reader to understand the scope and context of the risk assessment portion of your study.

Line 22. Same as above.  I cannot evaluate the relevance of not exceeding the WHO DALY threshold without knowing more about the analysis, including what pathogens this analysis is based on and the assumed exposures.  I also find the comparison to the WHO threshold surprising given the location (i.e., in the USA) of the study.

Line 40 -42.  Can you please provide a bit more specificity about “with mandatory pre-treatment that will provide a potable quality product prior to injection into non-potable zones of the Floridan Aquifer”?  Is this to mean that the surface water is treated with “drinking water treatment, including disinfection” and then injected to the aquifer– or something slightly different?  I will also note that this statement seems to be in conflict with the text on lines 58-61 – “Studies in other MAR and ASR systems have shown microbial indicators of human fecal contamination (i.e., total and fecal coliforms) and opportunistic pathogens (e.g., E.coli, Pseudomonas aeruginosa) survive the treatment processes prior to injection, thereby placing the ASR facility in violation of the coliform/E. coli regulatory standards within the SDWA” – Please clarify.

Line 76.  I am unclear why you would assess the health risks associated with drinking the recharge water if “the recovered water is not intended to be a potable source or product”.  It seems like those results could easily be taken out of context and misinterpreted.  The larger comment related to this is that I think the manuscript needs a clear explanation of the scope and context of the risk analysis early in the document so the reader can understand what is being done and why.  Several comments below stem from the lack of such a section.

Line 137 – 143.  The stated threshold of 1 infection per 10,000 people is not a threshold that is officially endorsed, sanctioned or even recommended by US EPA for drinking water.  The authors are correct that this benchmark level was set forth in citation 25, but it was done so as a suggested level of health protection by the authors of that citation, not the US EPA, and further, current drinking water regulations do not use it as a risk threshold. Refer to the Long Term 2 Enhanced Surface Water Treatment Rule, as an example illustrating this point.  This needs to be resolved throughout the manuscript.  The WHO DALY threshold is fine, however.  Finally, it seems like there should be a threshold for comparison to recreational water risks.  US EPA does provide a value for that in the 2012 RWQC document.  After reading the whole manuscript I understand why the recreational threshold is not discussed here. Nevertheless, please refer to my prior comment (Line 76) about scope and context.

Line 151-159.  I would highlight that this study does not include a representative and health protective surrogate of viral pathogen risks.  The PV1 enterovirus risk is not likely to be representative or conservatively health protective of the risk from other viral pathogens, especially those that cause most cases of infection from this type of exposure.  So, evaluation of the viral results relative to any risk based threshold will need to be conducted cautiously and caveated appropriately. 

Lines 161-167.  The explanation on lines 164-165 is very important for context to understand your study.  I suggest this be highlighted and be included in the abstract and the introduction. 

Lines 171-172. I suggest you explain why you are using point estimate values only in your assessment.  Why not do Monte Carlo simulation and include distributions as data are available for at least some of your parameters.

Table 3. I am confused about why E. coli is part of this assessment and how it is being used. On line 153 you indicate that this study includes the fecal indicator E. coli, which is not considered pathogenic and rather is an indicator of potential fecal contamination, to my knowledge.  Again, after reading the whole manuscript I see you are addressing pathogenic E. coli, but this should be explained earlier for clarity. 

Also, I am surprised at the use of point values for concentrations without any accounting for ranges, variability or uncertainty.  Please justify

However, I find the aim of the study (which I support) to be somewhat in conflict with the risk assessment approach that was used.  It seems to this reviewer, that this exposure assessment is fine –but the subsequent use of those data and comparison to drinking water risk benchmarks is confusing and somewhat arbitrary.  I would suggest you simply compute the daily recreational water risks, with and without inactivation, and discuss the risk reduction and the associated implications.

Line 183.  Does this formula also include the number of days prior to discharge?

Table 4.  Again, I am confused about why the indicator organisms E. coli is included here.  Also, why have the authors selected reference 34 for the dose response for Cryptosporidium rather than the updated Messner and Berger dose response relationship (these are the same authors that conducted the cited work)?

Line 192-194.   I find this section to be fundamentally problematic.  First, recreational water risks are evaluated on a per event basis (refer to US EPA 2012) with the benchmark set at an average risk of 32 or 36 illnesses per 1000 events.  There is no basis to compare recreational risks to an annual drinking water benchmark.  Second, what is the basis for using 30 exposures – this seems arbitrary.  Wouldn’t any comparisons to a benchmark will be influenced by this arbitrary decision? At a minimum this requires justification and a sensitivity analysis.

Line 214 -216.  This is the explanation regarding E. coli that I was missing earlier.  I suggest you move this explanation earlier in the text to the hazard identification section.

Section 3.  I suggest you critically rethink how you want to present your results.  I don’t understand the relevance of comparing the results to the drinking water benchmark, given the approach you have selected.  Specifically, 1) if you change the number of days of exposure, wouldn’t this change the comparison to the benchmark (as indicated, 30 days exposure for drinking water seems arbitrary)?; and 2) The comparison of recreational water exposure risks to a drinking water threshold is without basis.  It seems to me that the most relevant comparison would be single exposure events assuming the recreational exposure, as a worst case, compared to the average 32 illnesses / 1000 recreation event benchmark.  This could be done assuming various retention times (inactivation durations) and could support the perspective you describe on lines 291-299. 

Lines 299 – 304.  I believe that this argument could be made much stronger if the analysis were framed in a more coherent manner.  I understand and appreciate the main perspective, but unfortunately, I don’t believe that the analysis and results, as presented are compelling to support it.

Line 322 – 328.  Similar comment as above.  I find the arguments lacking given the assumption built into the assessment and the comparisons to benchmarks to be rather uninformative. 

Reviewer 3 Report

1. Abstract is not informative and add key findings of the research 

2. Lack of literature review in introduction section, add previous studies related to the aquifer depth and human health risk assessment 

3. Add study area description must be related to objectives of the study, but some information missing 

4. Use Arc GIS or QGIS for study area map representation, Add latitude and longitude of the study area

5. State the novelty of the present study in introduction section, i did not find any novel approach in methodology section 

6. Compare results with previous studies 

7. Revise conclusion and it should be comprise all the findings for study objectives 

Minor English correction required 

Reviewer 4 Report

Introduction well written. Materials and methods are described in detail and correctly. The results are presented very clearly and well discussed. The discussion of the results is correct. In the conclusion part, I suggest introducing some corrections. "The QMRA framework described in the study emphasizes the critical role that geochemical conditions in ASR storage have on pathogen decay, thereby mitigating associated human health risks without the implementation of expensive – and unnecessary – water treatment”. This sentence implies no need for water treatment. This view can be very misleading. If we look at the results of physical and chemical tests of water in Table 2, many parameters are wrong. It may also be wrong to look only at the microbiological aspect, although of course it is water microbiology that is considered in this model.

Suggests acceptance after minor corrections in the conclusions part.

Round 2

Reviewer 1 Report

The authors have addressed all the recommended modifications

Reviewer 3 Report

Check the subheadings and minor English correction required 

Spell check and minor english correction required